# Human ApoA-I Overexpression Enhances Macrophage-Specific Reverse Cholesterol Transport but Fails to Prevent Inherited Diabesity in Mice

**DOI:** 10.3390/ijms20030655

**Published:** 2019-02-02

**Authors:** Karen Alejandra Méndez-Lara, Núria Farré, David Santos, Andrea Rivas-Urbina, Jari Metso, José Luis Sánchez-Quesada, Vicenta Llorente-Cortes, Teresa L. Errico, Enrique Lerma, Matti Jauhiainen, Jesús M. Martín-Campos, Núria Alonso, Joan Carles Escolà-Gil, Francisco Blanco-Vaca, Josep Julve

**Affiliations:** 1Institut de Recerca de l’Hospital de la Santa Creu i Sant Pau i Institut d’Investigació Biomèdica Sant Pau, IIB-Sant Pau, 08025 Barcelona, Spain; kmendez@santpau.cat (K.A.M.-L.); nfarrec@santpau.cat (N.F.); arivas@santpau.cat (A.R.-U.); jsanchezq@santpau.cat (J.L.S.-Q.); terrico@santpau.cat (T.L.E.); jmartinca@santpau.cat (J.M.M.-C.); jescola@santpau.cat (J.C.E.-G.); 2Departament de Bioquímica i Biologia Molecular, Universitat Autònoma de Barcelona, 08193 Barcelona, Spain; 3CIBER de Diabetes y Enfermedades Metabólicas Asociadas, CIBERDEM, 28029 Madrid, Spain; daymer11@hotmail.com (D.S.); nalonso32416@yahoo.es (N.A.); 4Minerva Foundation Institute for Medical Research, Biomedicum 2U and National Institute for Health and Welfare, Genomics and Biomarkers Unit, FIN-00290 Helsinki, Finland; jari.metso@thl.fi (J.M.); matti.jauhiainen@thl.fi (M.J.); 5CSIC-ICCC-IIB-Sant Pau i CSIC-Institut d’Investigacions Biomèdiques de Barcelona (IIBB), 08025 Barcelona, Spain; cllorente@santpau.cat; 6CIBER de Enfermedades Cardiovasculares, CIBERCV, 28029 Madrid, Spain; 7Servei de Bioquímica, Hospital de la Santa Creu i Sant Pau i Institut d’Investigació Biomèdica de l’Hospital de la Santa Creu i Sant Pau, IIB-Sant Pau, 08041 Barcelona, Spain; elerma@santpau.cat; 8Departament de Patologia, Hospital de la Santa Creu i Sant Pau, 08041 Barcelona, Spain; 9Departament de Ciències Morfològiques, Universitat Autònoma de Barcelona, 08193 Bellaterra, Spain; 10Servei d’Endocrinologia, Hospital Universitari Germans Trias i Pujol, Badalona, 08916 Barcelona, Spain

**Keywords:** HDL functions, reverse cholesterol transport, metabolic syndrome, obesity, hepatic steatosis

## Abstract

Human apolipoprotein A-I (hApoA-I) overexpression improves high-density lipoprotein (HDL) function and the metabolic complications of obesity. We used a mouse model of diabesity, the db/db mouse, to examine the effects of hApoA-I on the two main functional properties of HDL, i.e., macrophage-specific reverse cholesterol transport (m-RCT) in vivo and the antioxidant potential, as well as the phenotypic features of obesity. HApoA-I transgenic (hA-I) mice were bred with nonobese control (db/+) mice to generate hApoA-I-overexpressing db/+ offspring, which were subsequently bred to obtain hA-I-db/db mice. Overexpression of hApoA-I significantly increased weight gain and the incidence of fatty liver in db/db mice. Weight gain was mainly explained by the increased caloric intake of hA-I-db/db mice (>1.2-fold). Overexpression of hApoA-I also produced a mixed type of dyslipidemia in db/db mice. Despite these deleterious effects, the overexpression of hApoA-I partially restored m-RCT in db/db mice to levels similar to nonobese control mice. Moreover, HDL from hA-I-db/db mice also enhanced the protection against low-density lipoprotein (LDL) oxidation compared with HDL from db/db mice. In conclusion, overexpression of hApoA-I in db/db mice enhanced two main anti-atherogenic HDL properties while exacerbating weight gain and the fatty liver phenotype. These adverse metabolic side-effects were also observed in obese mice subjected to long-term HDL-based therapies in independent studies and might raise concerns regarding the use of hApoA-I-mediated therapy in obese humans.

## 1. Introduction

Obesity and type 2 diabetes mellitus are metabolic conditions that are frequently associated with an increased risk of cardiovascular disease (CVD) [1]. The functional characteristics of high-density lipoprotein (HDL) are distorted in subjects with obesity and diabetes [2], and the latter partially accounts for the increased CVD risk in obese subjects and patients with diabetes.

In vivo macrophage-specific reverse cholesterol transport (m-RCT) is considered one of the most important anti-atherogenic properties of HDL [3]. HDL also possesses potent antioxidant properties [4,5].

Previous data support a relationship between the main atheroprotective properties of HDL and its apolipoprotein A-I (ApoA-I) content [4,5]. Several experimental strategies aimed at elevating ApoA-I levels or using mimetic peptides proved effective at improving the main HDL atheroprotective functions [6]. Accordingly, ApoA-I overexpression is associated with favorable anti-atherogenic effects, as it reduces atherosclerosis in different mouse models [7,8] and in rabbits [9,10]. In addition to its atheroprotective actions, compelling evidence also shows a role for ApoA-I in preventing adiposity in mice [11], whereas its deficiency promotes fat gain [12]. The overexpression of human (h)ApoA-I also alleviates experimental fatty liver and endoplasmic reticulum (ER) stress [13,14], which are common manifestations of obesity and diabetes [15,16]. In particular, hepatic ER stress directly influences hepatic cholesterol metabolism [17], thereby contributing to the dysregulation of HDL metabolism and function [18,19,20]. 

Although the hypothesis that ApoA-I improves the atheroprotective properties of HDL [21,22,23] is well documented, the impact of its overexpression on these properties in an animal model of diabesity (db/db mice) with dysfunctional HDL is yet to be established. Thus, we addressed this question by testing the hypothesis that two of the most important anti-atherogenic properties attributed to HDL, which are compromised in db/db mice, would be improved following the overexpression of hApoA-I in db/db mice. Moreover, we also studied whether hApoA-I overexpression produced a concomitant amelioration of obesity-related complications observed in an experimental model of diabesity.

## 2. Results

### 2.1. Human ApoA-I Overexpression Promotes Weight Gain and Fatty Liver in db/db Mice

The total caloric intake and body weight were higher in db/db mice maintained on a regular chow diet (~1.3-fold and ~1.6-fold, *p* < 0.05, respectively) than in nonobese control mice (Table 1). A significantly greater weight gain was observed in hApoA-I transgenic (hA-I)-db/db mice (hA-I-db/db mice: 0.38 g/day; daily increase: ~1.3-fold, *p* < 0.05) than in age-matched db/db mice (db/db mice: 0.30 g/day) (Table 1 and Figure 1a), and this finding was mainly attributed to an increased food intake (Table 1). Consistent with these findings, the epididymal fat pad was also larger (~1.3-fold, *p* < 0.05) in hA-I-db/db mice than in db/db mice. Plasma glucose levels in hA-I-db/db mice were similar to db/db mice (Table 1 and Appendix A). The overexpression of hApoA-I in db/db mice did not improve the impaired glucose tolerance in db/db mice (Appendix A). We performed pair-feeding experiments in which both db/db and hA-I-db/db mice were fed the same amount of food based on the daily food consumption of a separate group of db/db mice (Figure 1b,c). Under these conditions, hA-I-db/db mice no longer gained weight more rapidly than db/db mice. Similarly, the increased epididymal fat mass of hA-I-db/db mice was also neutralized by pair feeding (Figure 1d).

The obese phenotype displayed by db/db mice was also characterized by concomitant hepatic steatosis, as revealed by significant increases in the total hepatic levels (>3-fold, *p* < 0.05) of cholesterol, triglycerides, and free fatty acids (FFA) (Table 1). Even higher hepatic total cholesterol, triglyceride, and FFA levels were detected in hA-I-db/db mice than in db/db mice (cholesterol: ~3.7-fold, *p* < 0.05; triglycerides: ~3.5-fold, *p* < 0.05; FFA: >3-fold, *p* < 0.05) (Table 1). The plasma levels of hepatic enzymes (i.e., alanine transaminase (ALT) and aspartate transaminase (AST)) showed a commensurate increase (~2-fold, *p* < 0.05) in hA-I-db/db mice compared with db/db mice (Table 1). The hepatic triglyceride content was directly related to the plasma levels of transaminases (Appendix A), as well as liver and body weights (Appendix A).

The accumulation of fat in hepatocytes was further studied using liver histology. Hematoxylin and eosin staining revealed the presence of numerous large vacuoles that filled the cytoplasm and displaced the nucleus in hA-I-db/db mice compared with db/db mice. In contrast, no signs of steatosis were observed in either nonobese hA-I or control mice (Appendix A). Hepatic steatosis in db/db mice was predominantly characterized by the presence of lipid vacuoles of varying diameters, a hepatic phenotype also known as microvesicular steatosis (Appendix A). In contrast, liver specimens from hA-I-db/db mice predominantly showed the presence of lipid macrovesicles (Appendix A).

The increase in liver triglyceride levels in db/db mice was accompanied by a concomitant induction of the expression of messenger RNAs (mRNAs) encoding markers of hepatic FFA uptake (*Cd36*: ~5-fold, *p* < 0.05) and synthesis (*Acaca*: >2-fold, *p* < 0.05) (Table 2). The expression of the *Pparg* gene (~2-fold, *p* < 0.05), which specifically targets *Cd36* mRNA expression [24], was also upregulated in the livers of db/db mice (Table 2). Interestingly, the relative hepatic expression of *Acaca* and *Cd36* directly correlated with the hepatic triglyceride levels (*Acaca*: Spearman’s r = +0.59; *p* < 0.01; *Cd36*: Spearman’s r = +0.72; *p* < 0.01) (Appendix A). In contrast, lower expression of the *Hmgcr* gene, which is a main determinant of the de novo synthesis of cholesterol, was observed (>0.3-fold, *p* < 0.05) in the livers of db/db mice than in the livers of the nonobese control mice (Table 2).

Consistent with a previous report [25], the expression of the *Hspa5* gene, a known marker of ER stress, was significantly downregulated (~0.2-fold, *p* < 0.05) in db/db mice compared with nonobese control mice. Interestingly, our data also revealed that overexpression of hApoA-I in db/db mice failed to induce *Hspa5* expression (Table 2).

### 2.2. HApoA-I Overexpression Induces Dyslipidemia in Mice

Similar to a previous report [24], higher plasma total cholesterol levels (~+2-fold, *p* < 0.05) were observed in db/db mice than in nonobese control mice, mostly due to increased plasma levels of HDL and, to a lesser extent, non-HDL cholesterol (Table 1). The overexpression of hApoA-I in hA-I-db/db mice further increased the plasma levels of total cholesterol (~2.8-fold, *p* < 0.01), mostly due to elevations in the non-HDL fraction (~9.8-fold, *p* < 0.05), and triglycerides (~3.3-fold, *p* < 0.05) compared with db/db mice (Table 1). Although plasma levels of HDL cholesterol were elevated, the concentrations of mouse ApoA-I did not differ between db/db mice and nonobese control mice (Table 1), although hepatic expression of the mouse ApoA-I gene was dramatically reduced (>0.4-fold, *p* < 0.05) in db/db mice (Table 3). The plasma levels of the transgenic protein were elevated (~1.3-fold, *p* < 0.05) in hA-I-db/db mice compared with hA-I mice, whereas hepatic expression of the *APOA1* gene was substantially downregulated (>0.3-fold, *p* < 0.05) in hA-I-db/db mice (Table 3). While most of the circulating levels of hApoA-I were present in the HDL fraction in hA-I mice, only ~70% of this protein was detected in the HDL fraction of hA-I-db/db mice (Table 1). A significant amount, 30%, of the circulating hApoA-I protein was associated with the non-HDL fraction. Control mouse plasma HDL consisted of a single-sized population with a mean diameter of approximately 10 nm, whereas a smear of larger HDL coexisted with the 10-nm-sized HDL in db/db mice (Appendix A). As expected [26,27], the groups of mice overexpressing hApoA-I did not present detectable levels of the mouse form of the protein (Table 1) and presented a polydisperse pattern of HDL, which consisted of distinct HDL subpopulations (Appendix A). The overexpression of hApoA-I in db/db mice led to the appearance of larger and lower migrating species of HDL than the forms detected in hA-I mice. 

### 2.3. HApoA-I Overexpression Partially Restores the m-RCT in db/db Mice

The relative levels of [^3^H]-cholesterol in the plasma and HDL fraction were increased 1.5-fold (*p* < 0.05) in db/db mice compared with nonobese control mice (Figure 2a). The overexpression of hApoA-I in nonobese, control mice also increased the plasma levels of [^3^H]-cholesterol, which was primarily attributed to the increase in the HDL fraction (~3.3-fold, *p* < 0.05) (Figure 2a,b). As expected [21], the relative ability of hApoA-I-containing HDL to accept [^3^H]-cholesterol over 48 h was significantly increased (>3.5-fold) in the nonobese mice expressing hApoA-I (hA-I mice). Interestingly, although both db/db and hA-I mice showed similar plasma levels of HDL cholesterol (Table 1), significantly higher relative levels of [^3^H]-cholesterol (~2-fold) were detected in the hApoA-I-containing HDL than in HDL from db/db mice (Figure 2b). Furthermore, the relative increase in HDL-associated [^3^H]-cholesterol levels in hA-I-db/db mice was roughly for the same as the sum of the levels in db/db (~1.5-fold, *p* < 0.05) and hA-I mice (~3-fold, *p* < 0.05) (Figure 2b).

The analysis of the distribution of the [^3^H]-tracer in different compartments also revealed that both db/db and hA-I mice tended to accumulate higher levels of the [^3^H]-tracer in the liver than nonobese control mice (db/db mice: ~1.7-fold, *p* < 0.05; hA-I: ~2.1-fold, *p* < 0.05) (Figure 2c; Table 4). Notably, the elevated levels of hepatic [^3^H]-cholesterol in hA-I mice were not accompanied by changes in the expression of the *Abcg5/g8* genes (Table 3), suggesting that hepatic uptake increased, rather than decreased output of [^3^H]-cholesterol in these mice. Overexpression of hApoA-I in the hA-I-db/db mice further increased the hepatic accumulation of the [^3^H]-tracer compared with db/db mice (~2.9-fold, *p* < 0.05) (Figure 2c; Table 4). Interestingly, the relative hepatic levels of the [^3^H]-tracer detected in hA-I-db/db mice displayed an additive pattern, corresponding to the sum of the individual relative levels determined in both hA-I and db/db mice. Consistent with these findings, the relative hepatic gene expression of main transporters that control cholesterol trafficking, including *Abcg5/g8*, which controls the hepatobiliary flux of cholesterol, as well as *Abca1* and *Abcg1*, which control the cholesterol efflux from the liver into plasma, was significantly downregulated (>0.6-fold, *p* < 0.05) in both db/db and hA-I-db/db mice compared with control mice (Table 3), thereby explaining the cholesterol retention in the livers of hA-I-db/db mice.

Compared with nonobese control mice, the db/db mice showed a reduced fecal output of the [^3^H]-tracer (both in the form of cholesterol and bile acids) (~0.5-fold, *p* < 0.05) (Figure 3; Table 4). This finding confirmed previous data from our laboratory [24] indicating that the hepatobiliary trafficking of cholesterol into feces is impaired in db/db mice. As expected [21], overexpression of hApoA-I in the nonobese control hA-I mice increased the fecal levels of the [^3^H]-tracer (either in the form of cholesterol or biliary acids) (cholesterol: ~1.7-fold, *p* < 0.05; bile acid: ~1.5-fold, *p* < 0.05) (Figure 3; Table 4). Similarly, higher fecal release of the [^3^H]-tracer was also observed in hA-I-db/db mice than in db/db mice (~2.0-fold, *p* < 0.05) (Figure 3; Table 4).

In addition to ApoA-I, other known HDL-associated antioxidant proteins involved in mediating the atheroprotective functions of HDL were also analyzed. Total plasma levels of HDL-associated antioxidant enzymes (lipoprotein-associated phospholipase A2 (Lp-PLA2) and paraoxonase (PON1)) were elevated in db/db mice (~1.5-fold and ~1.3-fold, respectively; *p* < 0.05) compared with nonobese control mice (Table 5). The activity of Lp-PLA2 remained elevated in the plasma of hA-I-db/db mice compared with nonobese mice. However, the plasma levels of PON1 were decreased (>0.5-fold, *p* < 0.05) in both hA-I and hA-I-db/db mice compared with nonobese control and db/db mice. The observed differences in PON-1 and Lp-PLA2 activities were not explained by concomitant changes in the hepatic expression of their mRNAs (Table 5).

## 3. Discussion

For decades, overexpression of hApoA-I was reported to induce the main atheroprotective properties attributed to HDL, including m-RCT and antioxidant ability, and its clinical potential was assessed in animal and human models (reviewed in References [3,28]). To date, apoA-I was not reported to exert an effect on either body weight or food intake in mice with obesity induced by a high-fat diet [11,29]. Instead, it increases energy expenditure and reduces the fat mass in hA-I mice [11]. Consistent with the hApoA-I-mediated reduction in the fat mass, a number of independent studies also provided evidence for a protective role of hApoA-I in animal models of diet-induced fatty liver [30,31,32]. However, to our knowledge, no published studies evaluated the impact of hApoA-I overexpression on the abnormal functions of HDL in an animal model of diabesity (db/db mice). In the present study, overexpression of hApoA-I in db/db mice unexpectedly promoted weight gain and exacerbated the fatty liver phenotype, despite the improvement in m-RCT.

The exacerbations of weight gain and fatty liver produced by the overexpression of hApoA-I in db/db mice were associated with an elevated daily caloric intake. Our study is not the first to show deleterious metabolic outcomes induced by long-term therapies designed to improve HDL concentrations or functions. Consistent with our findings, other experimental strategies (i.e., microRNA 33 (miR-33) antagonism and cholesteryl ester transfer protein (CETP) inhibition) aimed at increasing circulating HDL levels and inducing the m-RCT were also reported to exert long-term, adverse metabolic effects on obese mice [33,34,35], including increased weight gain [35], dyslipidemia [34], and steatosis [33,34].

Overexpression of hApoA-I failed to prevent the fatty liver phenotype observed in db/db mice. The driving force underlying this failure was increased food intake, as steatosis was prevented by pair feeding with db/db mice. Fatty liver was mechanistically confirmed by a commensurate upregulation of lipogenic genes (i.e., *Acac*, and *Pparg*) in the livers of db/db mice. *Pparg* is generally activated in mouse models of diet-induced fatty liver [36,37]. Activation of *Pparg* was revealed by the hepatic upregulation of *Cd36*, a free fatty-acid uptake transporter and a known *Pparg*-target gene [38], in both db/db and hA-I-db/db mice. Consistent with this finding, *Cd36* upregulation was previously correlated with fatty liver development in different experimental settings and clinical studies [39,40,41,42]. Additional evidence for *Pparg* induction was provided by the hepatic downregulation of the expression of the *APOA1* gene in hA-I-db/db mice. The expression of the *APOA1* gene is downregulated by a specific *Pparg* agonist [43]. Similar to *APOA1*, *Apoa1* expression was also significantly downregulated in db/db mice compared with nonobese control mice. These findings were consistent with previous data showing similar transcriptional dysregulation of mouse ApoA-I in mice with defects in leptin signaling [44].

ER stress was identified as a crucial mechanism of fatty liver [45] that is commonly enhanced in the steatotic livers of obese mice deficient in leptin signaling, and it is induced by a high-fat diet [46,47,48]. The unfolded protein response (UPR), which is regarded as a complementary adaptive defense mechanism against ER stress, is regulated by specific ER-resident chaperones that protect cells from activating ER-stress-induced apoptosis and cell death. The 78-kDa glucose-regulated protein (Grp78) (encoded by the *Hspa5* gene), which is one of the main representatives of these chaperone proteins, plays a crucial role in modulating the UPR [49]. Consistent with previous observations [25], the hepatic expression of *Hspa5* was consistently downregulated in db/db mice compared with nonobese mice. We did not determine the hepatic levels of this ER chaperone protein, but experimental overexpression of the *Hspa5* gene directly relates to protein function, and independent studies described the overexpression of this gene as a favorable protective mechanism against ER stress in hepatocytes [49,50]. Thus, *Hspa5* downregulation in the livers of db/db mice would also be consistent with the enhanced development of hepatic steatosis in these mice and was not prevented in our hA-I-db/db mice.

Our hA-I-db/db mice also developed a mixed type of dyslipidemia, characterized in part by elevated plasma levels of triglycerides and non-HDL cholesterol. The mechanism underlying the accumulation of non-HDL remains unknown, but non-HDL accumulation might be promoted by increased plasma levels of hApoA-I [51]. HDL functions as apolipoprotein reservoir for the adequate maturation of triglyceride-rich lipoproteins [52]. Therefore, the excess hApoA-I in HDL might contribute to hypertriglyceridemia by altering the HDL proteome, thereby abrogating triglyceride-rich lipoproteins. A similar mechanism was in fact also described in mice overexpressing the second major apolipoprotein of HDL, hApoA-II, to explain the altered triglyceride-rich lipoprotein metabolism observed in these transgenic mice [53]. Alternatively, triglyceride-rich, non-HDL elevations might also be explained by an increased hepatic production. These mechanisms would likely be aggravated by the exacerbated hepatic steatosis and insulin resistance [54] displayed in mice on the db/db background. Further research is warranted to elucidate the negative impact of hApoA-I overexpression on diabesity in the db/db mouse.

We also used our mouse model to evaluate the impact of hApoA-I overexpression on m-RCT, which is likely one of the main anti-atherogenic properties of HDL [4]. Consistent with previous studies [24], the relative fecal levels of the [^3^H]-tracer (i.e., the last step in m-RCT) were reduced in db/db mice compared with nonobese control mice. As expected, overexpression of hApoA-I promoted the fecal excretion of the [^3^H]-tracer in nonobese control mice. Notably, fecal excretion of the [^3^H]-tracer was also stimulated in the hA-I-db/db mice and reached relative values similar to the nonobese control mice. The absence of changes in the analysis of the expression of the mRNAs encoding the main targets involved in cholesterol transport into the feces prompted us to hypothesize that the partial recovery of the relative fecal levels of the [^3^H]-tracer in hA-I-db/db mice was indirectly attributed to an increased hApoA-I-related upload of cholesterol in the livers of these mice. Indeed, these data would be consistent with previous data showing that upregulation of hApoA-I induced by fenofibrate promotes the hepatic accumulation of [^3^H]-cholesterol in male hA-I mice [55]. This hepatic accumulation of cholesterol may be potentially explained, at least in part, by the downregulation of genes (i.e., ATP-binding cassette (ABC) transporters) involved in the trafficking of hepatic cholesterol to the bile and plasma. The Abcg5/g8 complex requires simultaneous expression of both cholesterol transporters [56,57]. Consistent with a previous report [24], some liver X receptor (LXR) targets other than *Abcg5/g8*, such as *Cyp7a1* and *Cyp27a1*, were also downregulated in db/db mice, thereby suggesting the potential inhibition of LXR signaling in the livers of db/db mice [24]. In the present study, similar results were also obtained in hA-I-db/db mice, thus suggesting that this hepatic signaling pathway was also defective.

Another crucial atheroprotective action of HDL is to protect against low-density lipoprotein (LDL) oxidation, mainly by inhibiting the formation of toxic lipid hydroperoxides and promoting their removal from LDL [23]. In our study, HDL isolated from the plasma of db/db mice failed to protect LDL from oxidation compared with the HDL from nonobese control mice. This finding was first related to a relative reduction in the levels of antioxidant HDL-associated proteins (i.e., PON1, Lp-PLA2, and ApoA-I) in db/db mice. The overexpression of hApoA-I in db/db mice improved the antioxidant ability of HDL by preventing LDL oxidation ex vivo compared the HDL from db/db mice. This effect was not explained by favorable changes in the plasma levels of HDL-associated Lp-PLA2 and PON-1 activities. In fact, the plasma levels of PON1 were decreased in the groups of mice overexpressing hApoA-I. Similar to mouse ApoA-I [26,27], hApoA-I might conceivably compete with PON1 and displaces it from the HDL particles in hA-I-db/db mice. Therefore, based on our data, the enhanced antioxidant properties of the HDL from hA-I-db/db mice might be attributed to an increase in the relative hApoA-I content.

The present study has several limitations. Firstly, the observed changes in mRNA expression do not necessarily reflect changes in the protein content and function. Additionally, an analysis of the levels of the abovementioned cholesterol transporter proteins is warranted to confirm and extend our present observations. Finally, the mechanisms underlying the decreased activity of the LXR pathway and the increase in hepatic cholesterol accumulation were not explored. Thus, further studies are warranted.

## 4. Materials and Methods

### 4.1. Mice

All animal procedures were reviewed and approved by the Institutional Animal Care Committee and Use Committee of the Institut de Recerca de l’Hospital de la Santa Creu i Sant Pau and authorized by the Animal Experimental Committee of the local government (#9015) (1 July 2016) in accordance to the Spanish law (RD 53/2013) and European Directive 2010/63/EU, and the methods were conducted in accordance with the approved guidelines. Eight-week-old, nonobese, male and female mice heterozygous for Leprdb (db/+) on a C57BL/6J genetic background were obtained from Jackson Laboratories (Bar Harbor, ME; no. 000697). Obese males were obtained by breeding db/+ mice. The hApoA-I-overexpressing db/db mice (hA-I-db/db mice) were generated by first breeding nonobese control mice harboring the db/+ allele (db/+) with hApoA-I transgenic (hA-I) mice (Jackson Laboratories; no. 001927). The offspring harboring the db/+ allele that were positive for hApoA-I were further bred to obtain hApoA-I-overexpressing db/db (hA-I-db/db) and hApoA-I-overexpressing, nonobese db/+ (hA-I) mice. The hA-I offspring harboring the db/+ allele were then crossbred to obtain the number of mice required for the study. In parallel, hApoA-I negative mice were obtained by breeding db/+ mice. All mice (either lean or obese), including the hApoA-I positive and negative mice, were housed in a temperature-controlled environment (20 °C) with a 12-hour (h) light/dark cycle and maintained on a regular chow diet (Safe, Scientific Animal Food & Engineering; A04/A04C/R04, 3% fat, 2900 kcal/kg). Food and water were provided ad libitum. Genotyping was performed as indicated on the Jackson laboratory website (available online: http://jaxmice.jax.org). At the end of the studies, three-month-old mice were maintained ad libitum, euthanized, and exsanguinated by cardiac puncture. Blood was collected and serum was obtained by centrifugation. Food intake of age-matched, individually housed mice was monitored before (by preweighing a known amount of food) and after (by weighing the remaining food) consumption using wire-bottomed cages for 48 h before the animals were euthanized. In the pair-fed group, hA-I-db/db mice were fed the same amount of diet consumed by the db/db mice over the preceding 24 h. This amount of food was divided into two meals and provided at 8:00 a.m. and 6:00 p.m. to avoid long periods of fasting.

### 4.2. Laboratory Methods

#### 4.2.1. Biochemical Analyses

Plasma lipid, glucose, and transaminase analyses were performed enzymatically using commercial kits adapted for a COBAS c501 autoanalyzer (Roche Diagnostics SL, Sant Cugat del Vallès, Spain). Free cholesterol and phospholipids were determined using reagents from Wako Diagnostics (Wako Chemicals, Osaka, Japan). HDL cholesterol was measured in apoB-depleted plasma, obtained after precipitation with phosphotungstic acid and magnesium ions (Roche Diagnostics). HDL size was determined by non-denaturing 4–30% polyacrylamide gel gradient electrophoresis from plasma pre-stained with Sudan Black IV.

#### 4.2.2. Glucose Tolerance Test

Glucose tolerance tests were performed by administering an intraperitoneal injection of glucose (1 mg/g of body mass) after a 4-h fast. Plasma glucose levels were determined at *t* = 0 (baseline), 15, 60, 120, and 180 min. The area under the concentration curve (AUC) was calculated to compare glucose tolerance among groups [58].

#### 4.2.3. Measurement of m-RCT In Vivo

Macrophages (J774) were labeled with [^3^H]-cholesterol and intraperitoneally injected into mice, as previously described [59]. Mice were then individually housed in metabolic cages and stool samples were collected over a 48-h period. At 48 h, the mice were euthanized and plasma, liver tissues, and feces were collected and processed using the appropriate methods. Radioactivity was directly determined by liquid scintillation counting in plasma and HDL, which were obtained after precipitating non-HDL with phosphotungstic reagent. After extraction with isopropyl alcohol–hexane, the lipid layers of liver and fecal extracts were collected and evaporated, and the radioactivity of the [^3^H]-tracer was measured using liquid scintillation counting, whereas the content of the [^3^H]-tracer present in fecal bile acids was determined in the remaining aqueous phase of the fecal material extracts. The amount of [^3^H]-tracer was reported either as a fraction of the injected dose or as a relative value to the control group. The signal in control mice was set at a normalized value of 1 arbitrary unit.

#### 4.2.4. Fecal and Liver Lipid Analyses

Feces from individually housed mice that were provided access to feed and water ad libitum were collected over a 48-h period. Mice were euthanized and exsanguinated by cardiac puncture at the end of the study; the livers (100 mg) and feces (1 g) were collected, and lipids were extracted with isopropyl alcohol–hexane (2:3; *v*:*v*). The lipid layer of liver and fecal extracts was isolated after the addition of Na_2_SO_4_, evaporated with nitrogen, and emulsified with 0.5% sodium cholate prior to lipid measurements.

#### 4.2.5. Histology

Formalin-fixed liver tissues were embedded in paraffin and sectioned at a 5-μm thickness for hematoxylin and eosin (H&E) staining [60].

#### 4.2.6. Susceptibility to Lipoprotein Oxidation

Isolated lipoproteins were dialyzed in phosphate-buffered saline (PBS) by gel filtration on PD-10 columns (GE Healthcare, Barcelona, Spain). Oxidation was initiated by adding 2.5 μmol/L CuSO_4_ to human LDL (0.1 mmol/L phospholipids) and HDL (0.1 mmol/L of phospholipids) mixtures or to LDL and HDL alone [61]. The formation of conjugated dienes was measured by continuously monitoring the absorbance at 234 nm in a Synergy HT microplate reader (BioTek Synergy, Bad Friedrichshall, Germany) at 37 °C for 6 h. The lag phase was calculated as previously described [62]. Kinetics of LDL oxidation in the LDL + HDL incubations were calculated by subtracting the kinetics of HDL incubated without LDL.

#### 4.2.7. Analyses of HDL-Associated Proteins and Enzymes

The plasma levels of hApoA-I were determined using immunoturbidimetric commercial kits adapted to a COBAS c501 autoanalyzer (Roche Diagnostics). Mouse apoA-I levels were determined with an ELISA kit, and the wells were coated with a polyclonal rabbit antibody against mouse apoA-I, as previously reported [62]. Serum levels of arylesterase activity were determined using phenyl acetate as the substrate [60]. Ethylenediaminetetraacetic acid (EDTA)-sensitive serum arylesterase activity was calculated by subtracting the EDTA-resistant arylesterase activity in mixtures containing 1 mmol/L EDTA in the reaction buffer instead of calcium chloride (CaCl_2_) and reported as paraoxonase (PON)1 activity. Plasma levels of Lp-PLA2 activity were determined using 2-thio-PAF (Cayman Chemical, Ann Arbor, MI, USA) as the substrate, according to the manufacturer’s instructions.

#### 4.2.8. Preparation of Oxidized LDL

The study was approved by the Ethics Committee of the Hospital de la Santa Creu i Sant Pau in Barcelona. EDTA-treated plasma was obtained from normolipidemic volunteers. LDL (1.019–1.063 g/mL) was isolated by sequential ultracentrifugation at 100,000× *g* at 4 °C. Oxidized LDL (oxLDL) was obtained by incubating PBS-dialyzed LDL (0.1 mmol/L phospholipids) with 2.5 μM CuSO_4_ for 18 h at 37 °C [62]. The oxidation reaction was terminated by the addition of 0.5 mM EDTA and 2 μM butylated hydroxytoluene.

#### 4.2.9. Quantitative Real-Time PCR Analyses

Total RNA was extracted from both the liver and small intestine using TRIzol^®^ reagent (Life Technologies, Carlsbad, CA, USA), according to the manufacturer’s protocol, and purified using the RNeasy Plus Mini Kit (QIAGEN Iberia, Madrid, Spain). The integrities of the total RNA samples were determined using a bioanalyzer. Total RNA was reverse-transcribed with random oligo (dT)_23_ primers and a mixture of dNTPs (Sigma-Aldrich Quimica S.A., Madrid, Spain), M-ML reverse transcriptase, RNase H Minus, and Point Mutant (Promega Biotech Ibérica S.L., Madrid, Spain) to generate complementary DNA (cDNA) templates. The cDNAs were subjected to real-time PCR amplification using Taqman Master Mix (Applied Biosystems, Foster City, CA). Specific mouse Taqman probes (Applied Biosystems) were used for *Apoa1* (Mm00437569_m1), *Abca1* (Mm00442646_m1), *Abcg1* (Mm00437390_m1), *Abcg5* (Mm00446241_m1), *Abcg8* (Mm00445970_m1), *Abcb11* (Mm00446241_m1), *Acaca* (Mm01304257_m1), *Cd36* (Mm01135198_m1), *Cyp7a1* (Mm00484152_m1), *Cyp27a1* (Mm00470430_m1), *Hmgcr* (Mm01282499_m1), *Hspa5* (Mm00517691_m1), *Ncp1l1* (Mm01191972_m1), *Scarb1* (Mm00450236_m1), and *Gapdh* (Mm99999915_g1) (reference gene) to analyze gene expression in mouse tissues. Specific human Taqman probes (Applied Biosystems) were used for *APOA1* (Hs0163641_m1) and *ACTB* (Mm99999903_m1) (reference gene) to analyze the expression of these genes in mouse tissues. Real-time PCR assays were performed on a C1000 Thermal Cycler coupled to a CFX96 Real-Time System (Bio-Rad Laboratories SA, Madrid, Spain). All analyses were performed in duplicate. The relative mRNA expression levels were calculated using the ΔΔCt method.

### 4.3. Statistical Analysis

The data are presented as means ± SEM. Statistical analyses were performed using GraphPad Prism software (GPAD, version 5.0, San Diego, CA, USA). The effects of human apoA-I expression on gross and plasma chemical parameters, kinetic studies, HDL properties, or gene expression levels were determined using either a nonparametric Kruskal–Wallis test followed by the Dunn multiple comparison test or a parametric one-way ANOVA followed by the Newman–Keuls multiple comparison test, as appropriate. The relationships between hepatic gene expression and hepatic triglyceride levels or between weight gain and food intake were determined by calculating Pearson’s correlation coefficients. Differences between groups were considered statistically significant when the *p*-value was <0.05.

## 5. Conclusions

In conclusion, hApoA-I overexpression in db/db mice increased weight gain and hepatic steatosis, and produced a mixed type of hyperlipidemia, despite the improvements in the two main cardioprotective functions of HDL.

## Figures and Tables

**Figure 1 ijms-20-00655-f001:**
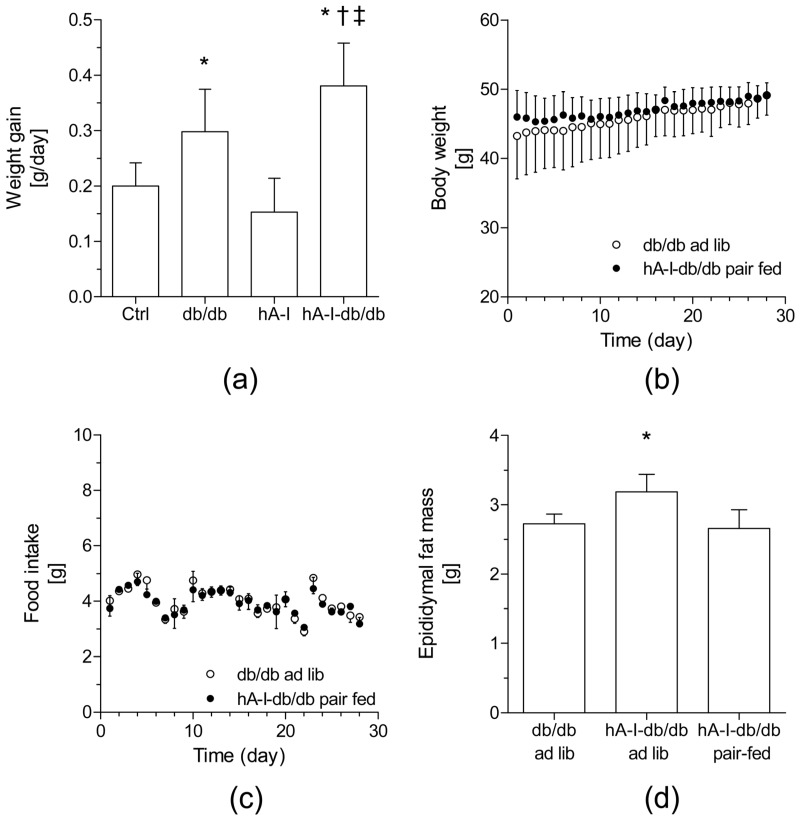
Overexpression of human apolipoprotein A-I (hApoA-I) increases weight gain in db/db mice. (**a**) Daily weight gain. The results are reported per gram of body weight. Means ± standard error values of data (*n* = 5–8 mice/group) are graphed. Differences between mean values were assessed using the nonparametric Kruskal–Wallis test followed by Dunn’s post-test or an ANOVA followed by the Newman–Keuls post-test, as appropriate; differences were considered significant at *p* < 0.05. Specifically, * *p* < 0.05 compared with the control (Ctrl) mice, † *p* < 0.05 compared with db/db mice, and ‡ *p* < 0.05 compared with hApoA-I transgenic (hA-I) mice. (**b**) Effects of hApoA-I overexpression or pair feeding on the body weight (**b**) and food intake (**c**) of db/db mice during the 28-day treatment period. (**d**) Measurements of the epididymal fat pad in pair-fed hA-I-db/db and db/db mice. In the pair-fed group, hA-I-db/db mice were fed the same amount of diet that was consumed ad libitum by db/db mice over the preceding 24 h. Values are presented as the means ± standard error of the mean (SEM) of 3 mice. † *p* < 0.05 compared with the pair-fed group (db/db mice).

**Figure 2 ijms-20-00655-f002:**
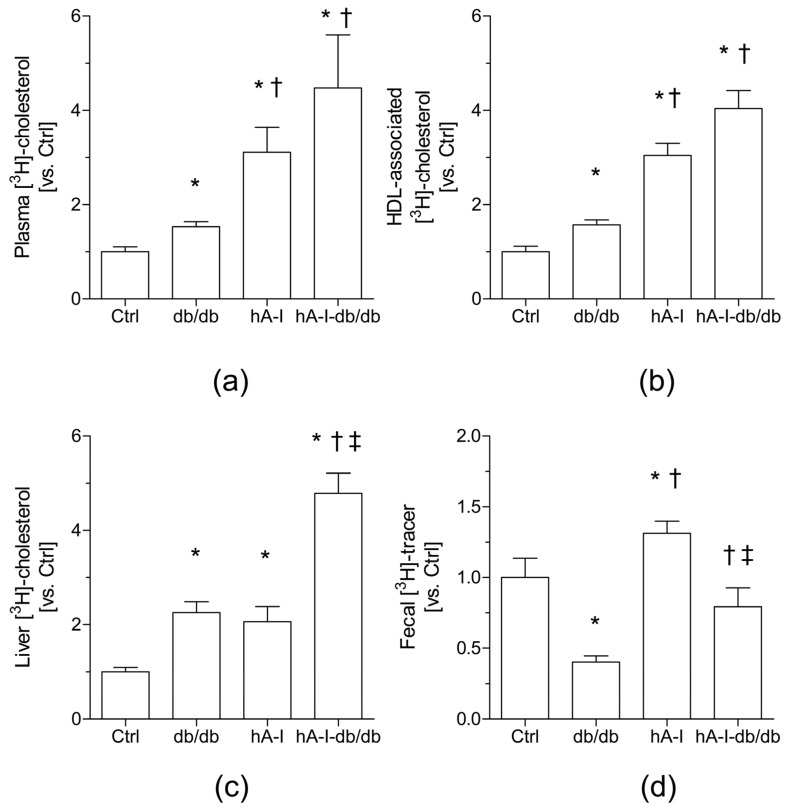
Overexpression of hApoA-I promotes macrophage-specific reverse cholesterol transport (m-RCT) in db/db mice in vivo. The macrophage-to-plasma m-RCT was increased in db/db mice, and the liver-to-feces m-RCT was partially restored in db/db mice overexpressing hApoA-I. Individually housed mice were intraperitoneally (i.p.) injected with [^3^H]-cholesterol-labeled J774 mouse macrophages, and the distribution of counts into different compartments was determined 48 h after the injection. Mice (either db/db mice or their littermates, i.e., Ctrl, and hA-I-db/db mice or their littermates, i.e., hA-I) were matched for body weight. In all panels, the results are presented as the means ± standard errors of 4–8 mice and are reported as a fold increase relative to the percentage observed in Ctrl mice compared to injected dose (vs. Ctrl). (**a**) Total plasma levels of [^3^H]-cholesterol (Ctrl: 1.00 ± 0.10% vs. the injected dose). (**b**) Plasma levels of [^3^H]-cholesterol in the high-density lipoprotein (HDL) fraction (Ctrl: 0.65 ± 0.04% vs. the injected dose). (**c**) Hepatic levels of [^3^H]-cholesterol (Ctrl: 1.87 ± 0.15% vs. the injected dose). (**d**) Fecal [^3^H]-tracer (Ctrl: 0.19 ± 0.02% vs. injected dose). Differences between mean values were assessed using the nonparametric Kruskal–Wallis test followed by Dunn’s post-test or an ANOVA followed by the Newman–Keuls post-test, as appropriate; differences were considered significant at *p* < 0.05. Specifically, * *p* < 0.05 compared with the control (Ctrl) mice, † *p* < 0.05 compared with db/db mice, and ‡ *p* < 0.05 compared with hApoA-I- transgenic mice (hA-I) mice. Abbreviations used: Ctrl, control mice; db/db, db/db mice; hA-I, hApoA-I-transgenic mice; hA-I-db/db, hApoA-I-overexpressing db/db mice.

**Figure 3 ijms-20-00655-f003:**
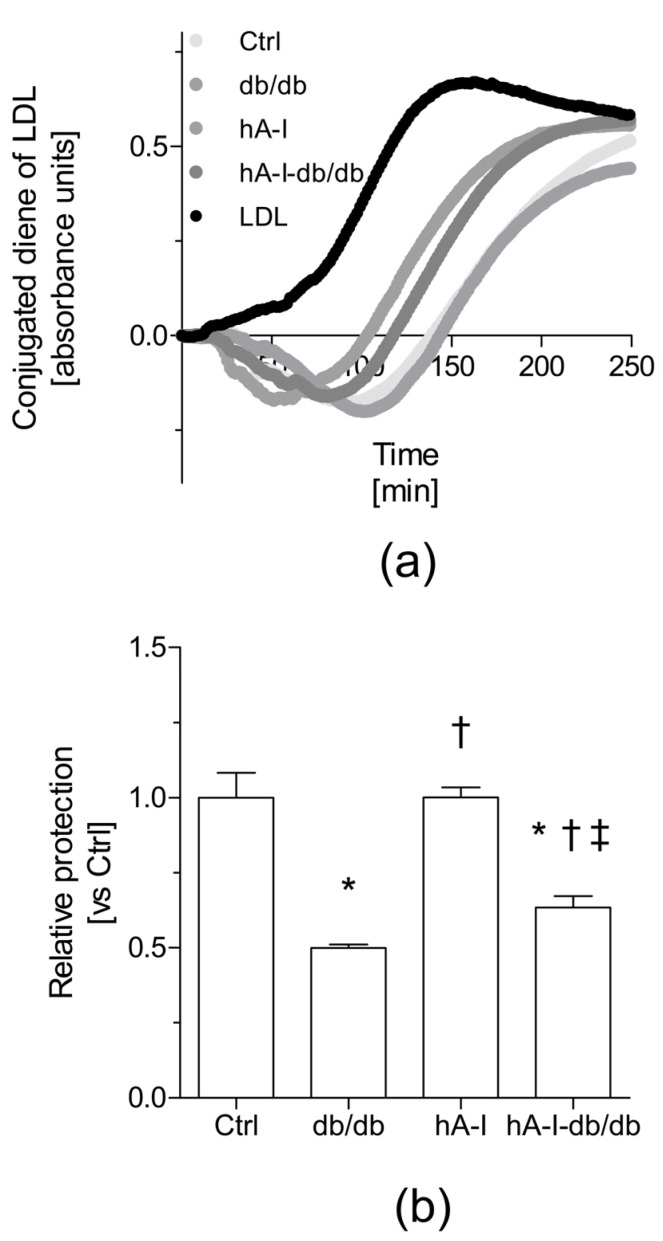
Overexpression of hApoA-I protects low-density lipoprotein (LDL) from ex vivo oxidation in db/db mice. The extent of copper-induced lipid oxidation was evaluated by measuring the formation of conjugated dienes in HDL + LDL mixtures. The results are reported as the lag phase of conjugated diene formation kinetics, which are presented as the relative lag phase to the kinetics of LDL oxidized without HDL (oxLDL). (**a**) Representative diene formation curves of human LDL co-incubated with HDL isolated from the different mouse groups in the presence of 2.5 μM Cu_2_SO_4_ at 37 °C. The kinetics of LDL oxidation are shown after subtracting the kinetics of HDL activity in the absence of LDL. (**b**) HDL antioxidant activity toward the LDL oxidative modification. For calculations, the mean lag time of oxLDL (52 min) was set to a normalized value of 1 arbitrary unit. The results are presented as the means ± standard errors of three independent pools of HDL obtained from 3–4 individual mice per group. Differences between mean values were assessed using the nonparametric Kruskal–Wallis test followed by Dunn’s post-test or an ANOVA followed by the Newman–Keuls post-test, as appropriate; differences were considered significant at *p* < 0.05. Specifically, * *p* < 0.05 compared with the Ctrl group, † *p* < 0.05 compared with db/db mice, and ‡ *p* < 0.05 compared with hA-I mice. Abbreviations used: Ctrl, control mice; db/db, db/db mice; hA-I, hApoA-I-transgenic mice; hA-I-db/db, hApoA-I-overexpressing db/db mice.

**Table 1 ijms-20-00655-t001:** Impact of human apolipoprotein A1 (hApoA-I) overexpression on gross and biochemical parameters in db/db mice.

Parameters	Ctrl	db/db	hA-I	hA-I-db/db	*p*
*Gross Parameterscharacteristics*					
Body weight (g)	28.7 ± 2.5	48.4 ± 0.8 *	29.4 ± 1.8 †	52.9 ± 1.6 * † ‡	<0.05
Liver weight (g)	1.4 ± 0.2	3.1 ± 0.1 *	1.5 ± 0.1 †	4.8 ± 0.3 * † ‡	<0.01
Food intake (kcal/day)	11.1 ± 0.1	14.8 ± 0.2 *	12.2 ± 0.2 †	18.6 ± 0.5 * † ‡	<0.01
Epididymal fat (g)	0.5 ± 0.1	2.7 ± 0.2 *	0.4 ± 0.1 †	3.5 ± 0.3 * † ‡	<0.01
*Plasma Biochemistry*					
Total cholesterol (mM)	2.6 ± 0.1	5.1 ± 0.4 *	4.8 ± 0.3 *	14.1 ± 0.8 * † ‡	<0.01
HDL cholesterol (mM)	2.1 ± 0.1	3.8 ± 0.3 *	4.2 ± 0.3 *	5.3 ± 0.6 * † ‡	<0.01
Non-HDL cholesterol (mM)	0.3 ± 0.0	0.9 ± 0.3 *	0.6 ± 0.1	8.8 ± 0.5 * † ‡	<0.01
Triglycerides (mM)	0.4 ± 0.0	0.6 ± 0.2	0.9 ± 0.1 *	2.0 ± 0.3 * ‡	<0.01
Total hApoA-I (g/L)	nd	nd	5.0 ± 0.3	6.5 ± 0.3 ‡	<0.05
HDL hApoA-I (g/L)	nd	nd	4.6 ± 0.6	4.5 ± 0.4	ns
Non-HDL hApoA-I (g/L)	nd	nd	0.2 ± 0.1	1.7 ± 0.2 ‡	<0.05
Mouse ApoA-I (g/L)	0.8 ± 0.1	1.1 ± 0.2	0.0 ± 0.0 * †	0.0 ± 0.0 * ‡	<0.01
Glucose (mM)	12.2 ± 0.6	17.2 ± 2.5 *	13.1 ± 0.6	17.6 ± 2.0 *	<0.01
*Liver-Related Biochemistry*					
ALT (U/L)	26 ± 2	95 ± 19 *	32 ± 3 †	231 ± 46 * † ‡	<0.01
AST (U/L)	65 ± 5	118 ± 21 *	86 ± 10	234 ± 20 * † ‡	<0.01
Cholesterol (µmol/liver)	5.3 ± 0.5	18.1 ± 4.4 *	8.6 ± 0.9 †	76.3 ± 10.5 * † ‡	<0.01
Triglycerides (µmol/liver)	5.1 ± 1.0	185.5 ± 24.1 *	13.6 ± 1.8 * †	661.3 ± 41.8 * † ‡	<0.01
FFA (µmol/liver)	22.6 ± 2.9	76.1 ± 6.3 *	46.0 ± 4.2 * †	256.9 ± 17.2 * † ‡	<0.01

Results are presented as means ± standard deviations (*n* = 5–8 mice per group). All analyses were performed at three months of age. Food intake was measured at the end of the study, as described in the Section 4. Plasma levels of the HDL fractions were determined in plasma supernatants after precipitation with phosphotungstic acid (Roche); the non-HDL fraction was calculated by subtracting the HDL moiety from the total plasma. Plasma levels of hApoA-I were determined using nephelometric commercial kits adapted to a COBAS c501 autoanalyzer. Mouse ApoA-I levels were determined using a specific ELISA kit, and the wells were coated with a polyclonal rabbit antibody against mouse apoA-I. Differences between the mean values were assessed using the nonparametric Kruskal–Wallis test followed by Dunn’s post-test or an ANOVA followed by the Newman–Keuls post-test, as appropriate; differences were considered significant at *p* < 0.05. Specifically, * *p* < 0.05 compared with the Ctrl group, † *p* < 0.05 compared with the db/db mice, and ‡ *p* < 0.05 compared with the hA-I mice. Abbreviations used: ALT, alanine transaminase; Apo, apolipoprotein; AST, aspartate transaminase; Ctrl, control mice; db/db, db/db mice; FFA, free fatty acids; hA-I, human apoA-I transgenic mice; hA-I-db/db, human apoA-I-overexpressing db/db mice; HDL, high-density lipoprotein; nd, not detectable; ns, nonsignificant.

**Table 2 ijms-20-00655-t002:** Hepatic gene expression profile of molecular targets involved in lipid metabolism and endoplasmic reticulum (ER) stress.

Genes	Ctrl	db/db	hA-I	hA-I-db/db	*p*
*Lipid Metabolism*					
*Acac*	1.0 ± 0.1	2.2 ± 0.3 *	0.8 ± 0.1 †	1.8 ± 0.3 * ‡	<0.01
*Cd36*	1.0 ± 0.1	5.0 ± 0.7 *	1.9 ± 0.6 †	4.7 ± 0.5 * ‡	<0.01
*Pparg*	1.0 ± 0.1	2.0 ± 0.3 *	1.0 ± 0.1	2.4 ± 0.4 * ‡	<0.01
*ER stress*					
*Hspa5*	1.0 ± 0.1	0.5 ± 0.1 *	0.7 ± 0.1	0.4 ± 0.1 *	<0.05

Results are presented as means ± standard deviations (*n* = 4–6 mice per group). The expression observed in Ctrl mice was set to a normalized value of 1 arbitrary unit. Differences between the mean values were assessed using the nonparametric Kruskal–Wallis test followed by Dunn’s post-test or an ANOVA followed by the Newman–Keuls post-test, as appropriate; differences were considered significant at *p* < 0.05. Specifically, * *p* < 0.05 compared with the untreated Ctrl group, † *p* < 0.05 compared with db/db mice, and ‡ *p* < 0.05 compared with db/db mice. Abbreviations used: Ctrl, control mice; db/db, db/db mice; hA-I, human apoA-I transgenic mice; hA-I-db/db, human apoA-I-overexpressing db/db mice; *Hspa5*, name of the gene encoding Grp78; ns, nonsignificant.

**Table 3 ijms-20-00655-t003:** Hepatic and intestinal gene expression profiles of molecular targets involved in macrophage-specific reverse cholesterol transport (m-RCT).

Genes	Ctrl	db/db	hA-I	hA-I-db/db	*p*
*Liver*					
*Apoa1*	1.0 ± 0.2	0.4 ± 0.1 *	0.6 ± 0.1	0.2 ± 0.1 * ‡	<0.01
*APOA1*	nd	nd	1.0 ± 0.1	0.3 ± 0.1 ‡	ns
*Abca1*	1.0 ± 0.1	0.4 ± 0.1 *	1.2 ± 0.1 †	0.4 ± 0.1 * ‡	<0.01
*Abcg1*	1.0 ± 0.1	0.3 ± 0.1 *	0.6 ± 0.1	0.4 ± 0.1 *	<0.05
*Abcg5*	1.0 ± 0.1	0.6 ± 0.1 *	0.9 ± 0.1 †	0.4 ± 0.1 * ‡	<0.01
*Abcg8*	1.0 ± 0.1	0.4 ± 0.1 *	0.9 ± 0.1 †	0.5 ± 0.1 * ‡	<0.01
*Scarb1*	1.0 ± 0.1	0.6 ± 0.1 *	1.1 ± 0.1 †	0.4 ± 0.0 * ‡	<0.01
*Abcb11*	1.0 ± 0.3	0.4 ± 0.2	0.6 ± 0.1	0.4 ± 0.1	ns
*Cyp7a1*	1.0 ± 0.2	0.5 ± 0.1 *	1.1 ± 0.2	0.2 ± 0.0 * ‡	<0.01
*Cyp27a1*	1.0 ± 0.1	0.6 ± 0.1 *	0.7 ± 0.1	0.2 ± 0.0 * ‡	<0.01
*Small intestine*					
*Abcg5*	1.0 ± 0.1	0.7 ± 0.1	0.7 ± 0.1	0.5 ± 0.1	ns
*Abcg8*	1.0 ± 0.1	0.9 ± 0.1	0.9 ± 0.1	0.8 ± 0.1	ns
*Ncp1l1*	1.0 ± 0.1	0.9 ± 0.2	0.6 ± 0.1	0.6 ± 0.1	ns

Results are presented as means ± standard deviations (*n* = 4–6 mice per group). The expression in Ctrl mice was set to a normalized value of 1 arbitrary unit. Differences between the mean values were assessed using the nonparametric Kruskal–Wallis test followed by Dunn’s post-test or an ANOVA followed by the Newman–Keuls post-test, as appropriate; differences were considered significant at *p* < 0.05. Specifically, * *p* < 0.05 compared with the untreated Ctrl group, † *p* < 0.05 compared with db/db mice, and ‡ *p* < 0.05 compared with db/db mice. Abbreviations used: Ctrl, control mice; db/db, db/db mice; hA-I, human apoA-I transgenic mice; hA-I-db/db, human apoA-I-overexpressing db/db mice; nd, not detectable; ns, nonsignificant.

**Table 4 ijms-20-00655-t004:** Fecal distribution of the [^3^H]-tracer in the cholesterol and bile-acid fractions in an m-RCT setting.

Parameters	Ctrl	db/db	hA-I	hA-I-db/db	*p*
Total activity	0.189 ± 0.021	0.083 ± 0.010 *	0.297 ± 0.020 * †	0.163 ± 0.032 † ‡	<0.01
Cholesterol	0.098 ± 0.013	0.040 ± 0.004 *	0.167 ± 0.010 * †	0.093 ± 0.019 † ‡	<0.01
Bile acid	0.088 ± 0.010	0.047 ± 0.005 *	0.130 ± 0.020 * †	0.073 ± 0.013 †	<0.01

Results are presented as means (percentage of the injected dose) ± standard deviations (*n* = 4–6 mice per group). Differences between the mean values were assessed using the nonparametric Kruskal–Wallis test followed by Dunn’s post-test or an ANOVA followed by the Newman–Keuls post-test, as appropriate; differences were considered significant at *p* < 0.05. Specifically, * *p* < 0.05 compared with the Ctrl group, † *p* < 0.05 compared with db/db mice, and ‡ *p* < 0.05 compared with hA-I mice. Abbreviations used: Ctrl, control mice; db/db, db/db mice; hA-I, human apoA-I transgenic mice; hA-I-db/db, human apoA-I-overexpressing db/db mice.

**Table 5 ijms-20-00655-t005:** Effect of human apoA-I overexpression on hepatic gene expression and plasma levels of antioxidant enzymes.

Parameters	Ctrl	db/db	hA-I	hA-I-db/db	*p*
*Gene Expression*					
*Pla2g7* (compared with the Ctrl)	1.0 ± 0.1	1.4 ± 0.3	1.3 ± 0.3	2.1 ± 0.4	ns
*Pon1* (compared with the Ctrl)	1.0 ± 0.1	0.6 ± 0.1 *	1.3 ± 0.2 †	0.5 ± 0.1 * ‡	<0.01
*Antioxidant Activities*					
Lp-PLA2 (µmol/mL/min)	134 ± 10	201 ± 14 *	104 ± 2 * †	227.2 ± 26 * ‡	<0.01
PON1 (µmol/mL/min)	56 ± 7	72 ± 3 *	29 ± 1 * †	33 ± 2 * †	<0.01

Results are presented as means ± standard deviations (*n* = 4–6 mice per group). Differences between the mean values were assessed using the nonparametric Kruskal–Wallis test followed by Dunn’s post-test or an ANOVA followed by the Newman–Keuls post-test, as appropriate; differences were considered significant at *p* < 0.05. Specifically, * *p* < 0.05 compared with the Ctrl group, † *p* < 0.05 compared with db/db mice, and ‡ *p* < 0.05 compared with hA-I mice. Abbreviations used: Ctrl, control mice; db/db, db/db mice; hA-I, human apoA-I transgenic mice; hA-I-db/db, human apoA-I-overexpressing db/db mice; *Pla2g7*, name of the gene encoding Lp-PLA2; Lp-PLA2, lipoprotein-associated phospholipase A2; PON1, paraoxonase.

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
