# Peer review of "Human ApoA-I Overexpression Enhances Macrophage-Specific Reverse Cholesterol Transport but Fails to Prevent Inherited Diabesity in Mice"

_ijms, 2019, doi:10.3390/ijms20030655_

Round 1

Reviewer 1 Report

The authors have clarified several of the questions regarding the antiatherogenic properties of HDL. The paper is well written, I recommend to accept it after minor revision:

line 28: "HDL" explanation of the abbreviation is listed in lines 49-50

line 65: "ER" stress - abbreviation should be explained

line 101: using a specific "an" ELISA kit

lines 103, 142, 229, 245, 264: nonparametric "a" Kruskal Wallis "test" (insertion "a" and missing "test")

line 309: genes genes (repeating words)

lines 311, 314, 316, 312?: "Pparg" should be formatted in an italic font

line 460: "real-time RT-" (repetition)

Author Response

Manuscript ID: ijms-421699

Title of the article Human ApoA-I overexpression enhances macrophage-specific reverse cholesterol transport but fails to prevent inherited diabesity in mice.

-Reviewer 1  -

The authors have clarified several of the questions regarding the antiatherogenic properties of HDL. The paper is well written, I recommend to accept it after minor revision:

We kindly thank the reviewer for the comments on the present study. We also acknowledge that some editing aspects have been improved in the revised version of the manuscript.

Therefore, we have properly corrected the following editing mistakes in the revised version of the manuscript:

line 28: "HDL" explanation of the abbreviation is listed in lines 49-50 Done

line 65: "ER" stress - abbreviation should be explained Done

line 101: using a specific "an" ELISA kit Done

lines 103, 142, 229, 245, 264: nonparametric "a" Kruskal Wallis "test" (insertion "a" and missing "test") Done

line 309: genes genes (repeating words) One of the two repeated terms has been removed from the text.

lines 311, 314, 316, 312?: "Pparg" should be formatted in an italic font

At the sites mentioned above, we were referring to the Pparg protein, not the gene; therefore, we decided to maintain the nonitalics notation of the transcription factor.

line 460: "real-time RT-" (repetition) Done

All suggested changes have been highlighted in the revised version of the manuscript.

Reviewer 2 Report

This manuscript looks at the physiological effects of overexpressing hApo-A1 in db/db mice. Authors have done extensive work on characterizing these effects and show that despite deleterious effects of weight gain and hepatic steatosis they do see increased RCT compared to db/db mice. This work warrants the publication in IJMS.  

Comments

Authors have extensively used the gene expression analysis to predict the functions of the probed protein. If authors could comment on would looking at protein expression of various transporters (abca1, abcg1, 5, and 8 along with SCARB1) involved in cholesterol transport change the conclusions authors have reached? The reason I ask this is that if the protein has long/high stability (low turnaround) in cells would RNA transcript levels go up?  E.g SCARB1 gene expression is low in hA-1-db/db mice compared to all other groups (table 3). Is this the case for SR-B1 protein levels as well? Some information on this area should be included in the discussion section.

Minor Comments

This work needs substantial English language correction before publication. Below are a few examples.

Line 53: Please Rephrase because the last part of the sentence does not make sense. Or remove the words ‘lipoproteins……. HDL’.

Line 58: Remove word ‘opinion’ since it is established using data in the literature, It would be good to use some other wording such ‘based on literature or data’.. or something on this line.

Line 74: should be ‘obesity-related complications.’

Author Response

This manuscript looks at the physiological effects of overexpressing hApo-A1 in db/db mice. Authors have done extensive work on characterizing these effects and show that despite deleterious effects of weight gain and hepatic steatosis they do see increased RCT compared to db/db mice. This work warrants the publication in IJMS.

Comments

Authors have extensively used the gene expression analysis to predict the functions of the probed protein. If authors could comment on would looking at protein expression of various transporters (abca1, abcg1, 5, and 8 along with SCARB1) involved in cholesterol transport change the conclusions authors have reached? The reason I ask this is that if the protein has long/high stability (low turnaround) in cells would RNA transcript levels go up?  E.g SCARB1 gene expression is low in hA-1-db/db mice compared to all other groups (table 3). Is this the case for SR-B1 protein levels as well? Some information on this area should be included in the discussion section.

We are grateful for the comments from reviewer 2. We agree with Reviewer 2 in that the observed changes in mRNA expression do not necessarily reflect changes in the protein content and function.

Consistent with a previous report (Errico, Mendez-Lara et al. 2017), db/db mice present impairments in hepatic LXR signaling, as revealed by the relative decrease in the levels of the Abca1, Abcg1, Abcg5/g8 mRNAs, which are known LXR targets. In addition to LXR signaling, these impairments might also be due to the impaired unfolded protein response (UPR) observed in these mice (Sabeva, Rouse et al. 2007). In particular, the expression of Grp78, an ER chaperone and component of the UPR, was downregulated and may further reduce the ER folding capacity of Abcg5/g8 and other cholesterol transporters, including SR-BI, thereby decreasing the levels and activities of these proteins.

Based on these findings, we hypothesized that the relative mRNA and protein levels of these cholesterol transporters would be reduced in db/db mice. Notably, our data further revealed that this phenotype was not prevented in our hApoA-I-overexpressing db/db mice; rather, it was found abrogated in these mice. The latter might be partially explained by the observation of larger amounts of cholesterol that accumulated in the livers of hA-I-db/db mice. Indeed, the hepatic accumulation of cholesterol was increased to a much greater extent in hA-I-db/db mice than in db/db mice, indicating that these mice might not be able to manage larger amounts of cholesterol.

In agreement with the comment from reviewer 2, we acknowledge that further studies analyzing the levels of the abovementioned cholesterol transporter proteins are warranted to confirm and extend our present observations in this mouse model. Therefore, the constraint mentioned by the reviewer has been added as a limitation of the study (discussion section, lines 393-398, pages 14-15).

Minor Comments

This work needs substantial English language correction before publication. Below are a few examples.

The English grammar and language have been reviewed by Scribendi Inc. (www.scribendi.com); we have used our best efforts to correct the English language in the revised version of the manuscript. This time English style and grammar have been reviewed by American Journal Experts.

Line 53: Please Rephrase because the last part of the sentence does not make sense. Or remove the words ‘lipoproteins……. HDL’. “Lipoproteins” and excess HDL notations have been removed from the text in the revised version of the manuscript.

Line 58: Remove word ‘opinion’ since it is established using data in the literature, It would be good to use some other wording such ‘based on literature or data’.. or something on this line.

“Previous data” have been added in place of “Current opinion”.

 Line 74: should be ‘obesity-related complications.’

This notation has been updated in the text in the revised version of the manuscript.